# Neurogenic Lower Urinary Tract Dysfunction in Asymptomatic Patients with Multiple Sclerosis

**DOI:** 10.3390/biomedicines10123260

**Published:** 2022-12-15

**Authors:** Anke K. Jaekel, Franziska I. Winterhagen, Federico L. Zeller, Anna-Lena Butscher, Franziska K. Knappe, Franziska Schmitz, Christopher Hauk, Johannes Stein, Ruth K. M. Kirschner-Hermanns, Stephanie C. Knüpfer

**Affiliations:** 1Department for Neuro-Urology, Clinic for Urology, University Hospital Bonn, 53127 Bonn, Germany; 2Neuro-Urology, Johanniter Neurological Rehabilitation Center ‘Godeshoehe e.V.’, 53177 Bonn, Germany; 3Clinic for Urology, University Hospital Bonn, 53127 Bonn, Germany

**Keywords:** multiple sclerosis (MS), neuro-urology, neurogenic lower urinary tract dysfunction (NLUTD), post-void residual (PVR), upper urinary tract damage (UUTD), bladder diary (BD), urodynamic study (UDS)

## Abstract

Neurogenic lower urinary tract dysfunction (NLUTD) in asymptomatic patients with MS has been described in preliminary studies, but specific investigations of this topic are rare. Many authors advise early diagnosis and treatment of NLUTD in patients with MS. In contrast, clinical practice and different guidelines recommend neuro-urological diagnostics only in the presence of symptoms. Our aim was to investigate the characteristics of NLUTD and the correlations of clinical parameters with NLUTD in asymptomatic patients with MS. We evaluated bladder diaries, urodynamic findings, and therapy proposals. Correlations of the voided volume, voiding frequency, urinary tract infections, and uroflowmetry including post-void residual with the urodynamic findings were determined. In our study, 26% of the patients were asymptomatic. Of these, 73.7% had urodynamic findings indicative of NLUTD, 21.1% had detrusor overactivity, 13.2% had detrusor underactivity, 13.2% detrusor overactivity and detrusor sphincter dyssynergia, and 57.9% had radiologically abnormal findings of the bladder. No patients presented low bladder compliance or renal reflux. Clinical parameters from the bladder diary and urinary tract infections were found to be correlated with NLUTD, and the absence of symptoms did not exclude NLUTD in patients with MS. We observed that urinary tract damage is already present in a relevant proportion. Based on our results, we recommend that patients with MS be screened for NLUTD regardless of the subjective presence of urinary symptoms or the disease stage.

## 1. Introduction

Although urinary symptoms are rare at first presentation of MS, up to 90% of patients experience lower urinary tract symptoms (LUTS) over the course of their disease [1]. Patients suffer mostly from detrusor overactivity (DO) (65%), detrusor underactivity/hypocontractility (DU) (25%), and detrusor sphincter dyssynergia (DSD) (35%) [2]. These pathophysiological alterations result in urgency, frequency, nocturia, and urinary incontinence or poor bladder emptying [1]. Neurogenic lower urinary tract symptoms (NLUTS) may be the most important socially disabling consequence of the condition [2]. 

Despite the serious impact on the affected patients, there is no uniform consensus on the optimal MS-specific urological diagnostics and therapy [1]. Although several European national panels have published their own guidelines for such patients, the guidelines are contradictory [1], and there is no uniform recommendation regarding referral for urodynamic studies (UDS) [3]. Some guidelines recommend further urological diagnosis only if symptoms and various complicating factors are present [4,5]. Other guidelines recommend that UDS should only be performed when treatment recommendations based on the baseline diagnostics of the symptomatic patient fail [6,7]. Asymptomatic patients are not considered in any of these reports, which ignores the fact that a relevant number of abnormal UDS findings have been reported in asymptomatic patients [8,9]. Furthermore, the absence of symptoms is not adequately reflected in UDS [10], and some guidelines recommend further neuro-urological examination regardless of the presence of NLUTS [9,11]. Neuro-urological diagnosis and therapy should be attained as early as possible to avoid secondary damage to the lower and upper urinary tract [3,11]. Therefore, we aimed to investigate the characteristics of neurogenic lower urinary tract dysfunction (NLUTD) in a group of urologically asymptomatic MS patients. Additionally, we assessed potential clinical predictors regarding their correlations with the UDS results indicative of NLUTD in this patient group.

## 2. Methods

### 2.1. Patients and Assessment

For this cross-sectional study, we included data of 256 patients with MS. Data were prospectively collected in the neuro-urological unit of an inpatient neurological rehabilitation center. For the present study, the database between February 2017 and July 2021 was analyzed retrospectively. The database consisted of patient and MS characteristics, urological history, data from bladder diaries, uroflowmetry (UF), UDS, and therapy recommendations. Inclusion criteria were age ≥18, definitive diagnosis of MS according to the McDonald criteria, the mental ability to answer questions, and written informed consent. Exclusion criteria were age <18 years, pregnant or breastfeeding, or an untreated acute lower UTI. 

The urological symptoms were evaluated by asking about bladder or micturition problems as described in the guidelines of the German Neurological Society for the diagnosis and treatment of MS patients [12]. Of the 256 patients, data on urologic symptomatology were available in 196 cases. The urinary tract infection rate (UTI) during the last 6 months was recorded by anamnesis. The data on mean voided volume (VVBD), minimal and maximal voided portion (MinVP, MaxVP), and voiding frequency (VF) per 24 h were obtained from a two-day bladder diary. The VF was standardized (SVF) according to a daily urine outtake of 2000 mL. The following formula was used [13]:$$\mathrm{SVF}=\frac{2000\;\mathrm{mL}}{\mathrm{voided}\;\mathrm{volume}\;\left(\mathrm{mL}/24\;h\right)}\times ø\mathrm{voiding}\mathrm{frequency}.$$

Uroflowmetry (UF) [14] with voided volume from uroflowmetry (VVUF) post-voided residual (PVR), and UDS were conducted according to the ICS standards [15]. The UDS were performed as videourodynamics (VUDS).

The correlation between the different clinical parameters and the pathological UDS indicative of NLUTD was assessed. The UDS findings indicative of NLUTD were defined by us, according to current doctrine [15] and according to a previous investigation [16]: first desire to void <100 mL, strong desire to void <250 mL, abnormal sensation or bladder capacity < 250 mL, bladder compliance < 20 mL/cm H_2_O, or any type of DO or DSD.

Furthermore, we assessed the correlation of these clinical parameters with the potential risk factors of upper urinary tract damage (UUTD) (DO and DSD, reduced compliance < 20 mL/cm H_2_O, and vesico-uretero-renal reflux (VUR)) as described by Ineichen et al. [3].

The clinical parameters and the thresholds were defined according to the results of a previous investigation [16] as follows, and MVP were additionally analyzed:
-Voided volume from bladder diary (VVBD)≤250 mL or ≥500 mL-Voided volume from uroflowmetry (VVUF)≥500 mL-Minimal voided portion (MinVP)≤50 mL-Maximal voided portion (MaxVP)≥500 mL-Urinary tract infection (UTI) rate>0/6 month-24 h standardized voiding frequency (SVF)≤4 or ≥13-Post-voided residual (PVR)>100 mL-Uroflowmetry (UF)abnormal curve or PVR > 100 mL or max flow rate < 10 mL/s

This study was conducted in accordance with the Declaration of Helsinki. All the patients gave written informed consent. Ethical approval (EK 313/13 University Hospital Bonn) was obtained. The database was registered in the German Clinical Trials Registry (DRKS 00024744).

### 2.2. Statistical Analysis

All analyses were performed with statistical programming language R (R Core Team 2019) (R version 4.1.0 (18 May 2021) on an x86_64-apple-darwin17.0 system running macOS Big Sur 10.16). To determine the correlations between the clinical parameters and the UDS findings, logistic regression models for the binary urologic findings were used to calculate the risks along with the expression of clinical parameters. Results with *p* < 0.05 were considered statistically significant.

## 3. Results 

### 3.1. Patient Characteristics, Clinical Parameters, and UDS Findings

In our study, 25.5% (50/196) of the patients were subjectively asymptomatic (no urinary complaints). Out of the asymptomatic patients, 42% (21/50) were male and 58% (29/50) were female. The patient characteristics are listed in Table 1. 

The clinical parameters were distributed as follows (Table 2).

Of the 50 asymptomatic patients, 12 cases did not undergo UDS. The analysis of the asymptomatic patients shows that 73.7% (28/38) of the UDS results are indicative of NLUTD. The distribution of the urodynamic results indicative of NLUTD is shown in Table 3. 

In our asymptomatic cohort, 21.1% (8/38) had DO, 13.2% (5/38) showed the simultaneous presence of DO and DSD, and 13.2% had DU (5/38). There were no asymptomatic patients with bladder compliance < 20 mL/cm H_2_O 0% (0/36). In the VUDS, 0% (0/38) had VUR, but 57.9% (22/38) had radiologically abnormal findings of the bladder: of these, 21.1% (8/38) had bladder diverticula, 13.2% (5/38) had descensus vesicae, 18.4% (7/38) had pathological changes of the bladder wall, and 21.1% (8/38) had pathological changes of the bladder neck or the proximal urethra. In our study, the UDS of asymptomatic patients resulted in treatment recommendations in 39.5% (15/38). In detail, intermittent self-catheterization (CIC) was recommended in 10.5% (4/38), antimuscarinics in 10.5% (4/38), alpha-blockers in 10.5% (4/38), physiotherapy of the pelvic floor and/or biofeedback therapy in 18.4% (7/38), and one case, 3% (1/38), received the recommendation of sacral neuromodulation.

### 3.2. Correlations between Clinical Parameters and UDS Findings Indicative of NLUTD or Risk Factors of Upper Urinary Tract Damage 

The clinical parameters VVBD ≤ 250 mL (relative risk (RR) 1.71, confidence interval (CI) 1.06–2.77, *p* = 0.027), VVBD ≥ 500 mL (RR 1.36, CI 1.04–1.78, *p* = 0.026), SVF ≥ 13/24 h (RR 1.38, CI 1.04–1.84, *p* = 0.026), MinVP ≤ 50 mL (RR 1.45, CI 1.05–2.02, *p* = 0.026), VVUF (RR 1.26, CI 1.05–1.52, *p* = 0.015), and UTI rate (RR 1.31, CI 1.09–1.58, *p* =0.005) showed significant correlation with the urodynamic findings indicative of NLUTD. The clinical parameters PVR, MaxVP ≥ 500 mL and abnormal UF did not show significant correlations with the UDS results indicative of NLUTD. No cases of reduced compliance < 20 mL/cm H_2_O or VUR were recorded in our asymptomatic cohort. Therefore, a correlation of the latter urodynamic results as potential risk factors of UUTD with clinical parameters could not be determined. We found no significant correlations between clinical parameters and DO and DSD. A summary of the correlations between the clinical parameters and the UDS findings indicative of NLUTD or risk factors of UUTD is given in Table 4.

## 4. Discussion

Symptoms of the lower urinary tract are frequent in patients with MS [1] and have a significant impact on their quality of life [17]. NLUTD is a main reason for morbidity and hospitalization [2], and yet NLUTD remains underdiagnosed (up to 48%) and undertreated [11]. 

It has been highlighted in previous studies that a significant proportion of asymptomatic patients with MS have NLUTD [8,9,13], and this absence of symptoms is not reflected in UDS [10]. Although a high number of pathologic UDS findings in clinically asymptomatic patients with MS was described as early as in 1991 by Bemelmanns et al. [8], there has been only one other specific study on this patient cohort [18].

This may be partly because NLUTD is considered to be of low importance in MS, as relevant upper urinary tract damage is rare [6,7,19]. From our point of view, the urological care and research of patients with MS seems to focus too much on the potential UUTD, and the NLUTD and the resulting damage to the lower urinary tract are not given enough weight. However, it is the symptoms of the lower urinary tract that limit the patient in everyday life [2,17]. They contribute to infection risk resulting in morbidity and hospitalization [9]. Moreover, there is evidence that the symptoms of NLUTD are also related to fatigue in patients with MS [20]. Fatigue is considered the most distressing symptom of MS and is difficult to treat [21]. Another explanation for the lack of studies on asymptomatic patients with MS lies in the patient cohort itself. Asymptomatic patients are not present for neuro-urological diagnostics. Even patients with LUTS present too infrequently, because the patients often do not provide information about LUTS, and physicians are unlikely to ask [22]. Therefore, systematic prospective and longitudinal studies with asymptomatic patients are difficult to realize. Our data come from an inpatient neurological rehabilitation center, where every patient with MS receives a neuro-urological presentation, which is why asymptomatic patients were examined in detail. 

Our examinations revealed 25.5% (50/196) of the patients were subjectively asymptomatic (no urinary complaints), and of these, 73.7% had UDS findings indicative of NLUTD. Other studies show variable findings on this point. While Seddone et al. [17] reported 47% and Bemelmanns et al. [8] reported 68% of urologically asymptomatic patients, Monti Bragadin et al. found that only 8.6% of patients were asymptomatic in their study [23]. The difference can be explained by the definition of “asymptomatic”. Monti Bragadin et al. used a LUTS-specific questionnaire for this purpose [23]. Thus, they used a diagnostic tool as recommended, e.g., in the Italian guidelines [24] and preselected the patient collective. Sedonne et al., in their cross-sectional retrospective study, checked for OAB symptoms and UTIs, feeling of incomplete emptying, hesitancy, or urinary retention [17]. In our study, we simply asked patients about bladder or urination problems. Our results showed that we would miss two-thirds of NLUTD in this patient cohort if we had based further diagnostics on this question. However, this corresponds most closely to the practice in neurological or general practitioner consultations [12]. Most MS guidelines and algorithms include the asymptomatic patient, but the definition of “asymptomatic” is diverse [4,5,7,9,11,13,18]. Without a clear definition, it is impossible to develop uniform algorithms for diagnostics and therapy in asymptomatic patients with MS. Therefore, this definition is mandatory prior to setting new algorithms.

One aim of our study was to investigate the correlations of the clinical parameters and NLUTD in asymptomatic patients with MS. Increased VVBD and VVUF of more than 500 mL and UTI rate > 0/6 months showed significant correlations with abnormal UDS within an asymptomatic cohort. This could be in line with the theory of Bemelmanns et al. regarding hyposensitivity and NLUTD in asymptomatic patients. They postulated a hyposensitive bladder as a reason for the high number of NLUTD in asymptomatic patients [8]. MS-related hyposensitivity of the bladder could lead to elevated levels of VV without causing symptoms. Due to its nature, hyposensitivity might be the most overseen diagnosis. A reduced VV ≤ 250 mL was predictive of the presence of NLUTD in the asymptomatic cohort. Likewise, asymptomatic individuals with MS with a small MinVP and a high voiding frequency show an increased risk for the presence of abnormal UDS in our study. A possible explanation would be that the urological symptomatology takes a back seat to the other symptoms such as paresis, depression, and fatigue. Moreover, even patients with MS who feel LUTS see their problem as too minor [22]. A bladder diary could help to objectify symptoms, regardless of the cause of the missing complaints. In various studies, VV has been used as a control in drug therapy [25,26]. Only one study found associations between OAB symptoms and VV, with a significantly higher VV when no frequency and urgency were reported [27], but only non-neurogenic patients were included. A threshold for VV as a predictor of NLUTD in asymptomatic individuals with MS has not yet been mentioned. 

Other parameters of the BD also lack clear thresholds for determining pathologic values. The interpretation is not standardized [18]. On the other hand, the diagnostic value of a BD regarding NLUTD has been confirmed in a variety of studies [4,5,9,24], although the resulting diagnostic steps are inconsistent [9,24], and clear recommendations based on the BD details are missing [28]. 

Furthermore, the guidelines emphasize that bladder diaries should be used in the presence of LUTS [4,5,9,24]. There was only one study to objectify missing symptoms, but it failed to identify a proper tool [18]. Our data, with 73.7% of asymptomatic patients having NLUTD resulting in 39.5% of therapy proposals, suggest that screening must start before symptoms occur. A bladder diary for every patient with MS could be an option to objectify the absence of symptoms. To determine their expressive power, it is necessary to conduct further investigations on the clinical predictors and thresholds obtained from the contents of patient bladder diaries. 

Finally, we must emphasize the question of which and how many asymptomatic patients with MS and NLUTD are not recognized in daily practice, and how this lack of recognition can be prevented. It is currently completely unclear whether asymptomatic MS patients whose NLUTD remains undiagnosed are or are not those whose treatment options are limited later in life due to massive trabeculation of the bladder wall or low bladder compliance resulting in permanent loss of the storage and voiding function of the lower urinary tract. In our study, we found that 57.9% of our patients had anatomical changes of the lower urinary tract in VUDS. We do not know if or when these patients will show symptoms. This has a special significance, as longer courses of MS are postulated to cause further pathological changes in the urinary tract over time [2,17], so early and targeted therapy is essential to avoid secondary damage [2,11,17]. To improve therapy response, knowledge-based decisions must be made, as the need to change therapy measures or failures of symptomatic treatment are negative predictors of therapy response [17]. Therefore, neurologists and general practitioners should cooperate as early as possible with urologists to evaluate MS patients at potential risk for developing NLUTD. The examination of patients in a neuro-urological MS center can allow impairments of the lower urinary tract to be detected at an early stage. Prospective and longitudinal studies are essential to develop uniform and systematic diagnostical and therapeutical algorithms for asymptomatic patients with MS.

## 5. Conclusions

The absence of urological symptoms does not exclude NLUTD in patients with MS. NLUTD was already present in 73.7% of an asymptomatic cohort. As VVBD ≤ 250 mL and ≥ 500 mL, VVUF ≥ 500 mL, SVF ≥ 13/24 h, MinVP ≤ 50 mL and lower UTI > 0/6 month are associated with NLUTD in this patient group, BD, uroflowmetry, and the history of UTI can help to objectify the absence of symptoms. Patients with MS should be examined for NLUTD regardless of the subjective presence of urinary symptoms and independent of the disease stage. Neurologists and general practitioners should cooperate as early as possible with urologists to evaluate MS patients at potential risk for developing NLUTD. Early screening for NLUTD enables knowledge-based therapeutical concepts. 

## 6. Limitations

One limitation was the low number of completed bladder diaries among asymptomatic patients with MS. Therefore, our results are based on a small data count in this section. This reflects the data situation of this patient cohort in general. Data were collected by a highly specialized neuro-urological department; thus, the recorded baseline characteristics of patients may deviate from those in less specialized settings. However, there are few data on asymptomatic patients with MS in urology outpatient clinics, because these patients are not routinely neuro-urologically evaluated. Another limitation is the lack of detrusor hypocontractility/underactivity in the definition of UDS results indicative of NLUTD. 

## Figures and Tables

**Table 1 biomedicines-10-03260-t001:** Characteristics of the asymptomatic patients with MS.

	Mean (SD)	Median (25–75%)	Min-Max	Missing % (N)
Age of patients in years	47.8 (10.71)	49 (40.25; 56.5)	18; 68	0% (0)
EDSS	3.83 (1.79)	3 (2.5; 5)	1.5; 8	42% (21)
Disease duration in months	9.04 (8.01)	7 (2.0; 14.5)	0; 32	2% (1)
MS Type	**% (*n*)**			
PPMS	10% (5)			2% (1)
RRMS	40% (20)			
SPMS	48% (24)			

N, number of patients; SD, standard deviation; EDSS, Expanded Disability Status Scale; MS, multiple sclerosis; PPMS, primary progressive MS; RRMS, relapsing remitting MS; SPMS, secondary progressive MS.

**Table 2 biomedicines-10-03260-t002:** Distribution of the clinical parameters in the asymptomatic patient cohort.

Clinical Parameter	Missing	N	Result
UTI 0/6 months	1	49	2.0% (1/49)
SVF ≤ 4/24 h	20	30	0.0% (0/30)
SVF ≥ 13/24 h	20	30	6.7% (2/30)
VVBD ≤ 250 mL	20	30	40.0% (12/30)
VVBD ≥ 500 mL	20	30	3.3% (1/30)
**MinVP** ≤ 50 mL	20	30	23.3% (7/30)
**MaxVP** ≥ 500 mL	20	30	46.7% (14/30)
Abnormal UF	4	46	39% (22/46)
**VVUF** ≥ 500 mL	4	46	17.4% (8/46)
PVR > 100 mL	4	46	21.7% (10/46)

N, number of patients; UTI, urinary tract infections; SVF, standardized voiding frequency; VVBD, mean voided volume from bladder diary; MinVP, minimal voided portion; MaxVP, maximal voided portion; VVUF, voided volume from uroflowmetry; UF, uroflowmetry; PVR, post-void residual.

**Table 3 biomedicines-10-03260-t003:** Urodynamic results indicative of NLUTD.

UDS Finding	Missing, UDS	Missing, Single Value	N	Result
First desire to void < 100 mL	12	0	38	10.5% (4/38)
Strong desire to void < 250 mL	12	5	33	24.2% (8/33)
Abnormal sensation	12	0	38	60.5% (23/38)
Bladder capacity < 250 mL	12	0	38	5.3% (2/38)
Compliance < 20 mL/cm H_2_O	12	2	36	0% (0/36)
Detrusor overactivity	12	0	38	21.1% (8/38)
Detrusor sphincter dyssynergia	12	3	35	22.9% (8/35)

UDS, urodynamic study; N, number of available data.

**Table 4 biomedicines-10-03260-t004:** Correlations between the clinical parameters and the UDS findings indicative of NLUTD or risk factors of UUTD.

	VVBD ≤ 250 mL	VVBD ≥ 500 mL	SVF ≥ 13/24 h	MinVP ≤ 50 mL	UTI > 0/6 Month	VVUF ≥ 500 mL
UDS findings indicative of NLUTD	RR 1.71 CI 1.06-2.77 *p* = 0.027	RR 1.36 CI 1.04-1.78 *p* = 0.026	RR 1.38 CI 1.04–1.84 *p* = 0.026	RR 1.45 CI 1.05–2.02 *p* = 0.026	RR 1.31 CI 1.09–1.58 *p* =0.005-	RR 1.26 CI 1.05–1.52 *p* = 0.015
DSD and DO	-	-	-	-	-	RR 1.45 CI 0.2–10.45 *p* = 0.712

UDS, urodynamic study; NLUTD neurogenic lower urinary tract dysfunction; DSD, detrusor sphincter dyssynergia; DO, detrusor overactivity; UTI, urinary tract infection; SVF, standardized voiding frequency; VVBD, mean voided volume from bladder diary; VVUF, voided volume from uroflowmetry; MinVP, minimal voided volume; RR, relative risk; CI, confidence interval

## Data Availability

The data presented in this study are available on request from the corresponding author. The data are not publicly available due to privacy.

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
