# Peer review of "Neurogenic Lower Urinary Tract Dysfunction in Asymptomatic Patients with Multiple Sclerosis"

_biomedicines, 2022, doi:10.3390/biomedicines10123260_

Round 1
Reviewer 1 Report
Very interesting paper, several comments as follows:
1. Abstract - spelling mistake:
and. 61% had radiologically abnormal findings of the bladder.
2. Materials and methods: we included data from 256 patients with MS -
Can authors explain data (patient) selection? Random 256 patients were included or 256 is the number of all patients evaluated 2017-2021 in the neuro-urological unit?
3. Results: In our study, 26% (50/196) were subjectively asymptomatic
It is listed that 256 patients are included, is it 196 or 256?
4. Results Table 2:
SVF ≤ 4 ×/24 h – can authors explain the high percentage of missing data?
5. Results Table 3. UDS Finding – in total 37 patients were evaluated out of 50? Can authors explain are 13 data missing or not performed?
Reviewer 2 Report
The study by Jaekel and coworkers focused on neurogenic lower urinary tract dysfunction (NLTUD) in patients suffering from multiple sclerosis. The auhtors found that a significant proportion of the patients included presented no urinary complaints. Still, despite lack of complaints, in this group of assymptomatic patients, urodynamic assessment showed that many presented altered values, indicative of NLTUD. Moreover, other patients presented other urodynamic findings, including detrusor over and underactivity. Finally, some patients also presented radiologic abnormalities.
While at some point most MS patients will present urinary complaints, urinary function is rearely evaluated at early time points of disease progression. In fact, MS patients are often only seen by specialized urologist when there is little to do to protect the urinary tract. Despited the enormous weight of urinary impairment in the quality of life, alleviative mesures are only offerered to MS patients at very late time points. Therefore, this study indicates that urological care should be offered much earlier than it currently is.
A few points should be improved for clarity sake:
- it seems the authors differentiate between NLTUD, detrusor overactivity and detrusor underactivity. For the non-specialists, NLTUD could incorporate both conditions. It would be helpful to clarify this point.
- were any differences between males and females? I ask this because table do not present that information.
- was there any correlation between the presence of NLTUD and the severity of other MS symptons and/or neurological lesions?
- what was the rationale behind the selected time frame?
